# A Familiar Outbreak of Monophasic *Salmonella* serovar Typhimurium (ST34) Involving Three Dogs and Their Owner’s Children

**DOI:** 10.3390/pathogens11121500

**Published:** 2022-12-08

**Authors:** Valeria Russini, Carlo Corradini, Emilia Rasile, Giuliana Terracciano, Matteo Senese, Federica Bellagamba, Roberta Amoruso, Francesco Bottoni, Paola De Santis, Stefano Bilei, Maria Laura De Marchis, Teresa Bossù

**Affiliations:** 1Istituto Zooprofilattico Sperimentale del Lazio e della Toscana “M. Aleandri”—Sezione di Roma, 00178 Rome, Italy; 2Istituto Zooprofilattico Sperimentale del Lazio e della Toscana “M. Aleandri”—UOT Toscana Nord, 56123 Pisa, Italy; 3National Reference Laboratory for Antimicrobial Resistance, General Diagnostics Department, Istituto Zooprofilattico Sperimentale del Lazio e della Toscana “M. Aleandri”, 00178 Rome, Italy

**Keywords:** *Salmonella*, One Health, non-typhoidal *Salmonella*, outbreak, ST34, infants, multidrug-resistance, pets

## Abstract

*Salmonella* is a Gram-negative enteric bacterium responsible for the foodborne and waterborne disease salmonellosis, which is the second most reported bacterial zoonosis in humans. Many animals are potential sources of salmonellosis, including dogs, cats, and other pets. We report the case of an outbreak of salmonellosis in a family in central Italy, affecting two children and involving their three dogs as carriers. One of the children needed medical care and hospitalisation. Isolation and analysis of stool samples from the sibling and the animals present in the house were carried out. Serotyping allowed the identification of *S. enterica* subsp. *enterica* serovar Typhimurium in its monophasic variant for all the isolates. The results of whole-genome sequencing confirmed that the strains were tightly related. The minimum inhibitory concentration (MIC) test documented the resistance to ampicillin, sulfamethoxazole, and tetracycline. The origin of the zoonotic outbreak could not be assessed; however, the case study showed a clear passage of the pathogen between the human and non-human members of the family. The possibility of a transmission from a dog to a human suggests the need for further studies on the potential ways of transmission of salmonellosis through standard and alternative feed.

## 1. Introduction

*Salmonella* is a Gram-negative bacterium responsible for the enteric foodborne and waterborne disease called salmonellosis, which was the second most reported bacterial zoonosis in humans in 2020: 52,702 human salmonellosis cases were reported by 27 European Union Member States (EU MS) in 2020, with an EU notification rate of 13.7 cases per 100,000 population. *Salmonella* spp. is the most frequently reported causative agent for foodborne outbreaks. Besides a decrease in cases observed in 2020, probably due to the effects of the COVID-19 pandemic, the overall trend for salmonellosis in the last five years did not show any statistically significant change. *S.* Enteritidis caused the majority (57.9%) of the reported foodborne outbreaks of *Salmonella* in the EU, vehiculated mostly by the consumption of “eggs and egg products”, “pig meat and product thereof”, and “bakery products” [1].

The highest notification rate of salmonellosis was observed among children of 0–4 years, with 94.1 cases per 100,000 population. The rate in young children was almost three times higher than in older children and eight times as high as in adults (25–64 years old) in 2017 [2].

In Italy, salmonellosis is the most reported foodborne zoonosis (3768 confirmed cases in 2021), with a rate of infections per 100,000 population significantly lower than most of the European rates (6.36 in 2021) [3]. Even if salmonellosis is considered a foodborne disease, direct and indirect contact with animals is a potential source of infection, including dogs and cats [4], cattle, horses, domestic rodents and non-traditional mammalian pets such as hedgehogs, sugar gliders and wallabies [5], reptiles, amphibians [6], and even home aquariums [7]. Reverse zoonotic disease (from human to animal, also defined as zooanthroponosis) is a seldom-documented way of transmission that should be considered in the study of zoonoses. This gap in scientific literature has been reported in order to increase awareness of the potential health threats inflicted on a susceptible animal by an unhealthy human and the subsequent transmission from the animal to other humans [8].

Herein, the case of a familiar salmonellosis outbreak is presented, which occurred in central Italy, involving two children and three of their four dogs. The causative strain of the outbreak was found to belong to the *S.* Typhimurium serovar in its monophasic variant, one of the most relevant for public health. Through the use of whole-genome sequencing techniques, it was possible to highlight the genetic link present between human and animal *Salmonella* isolates. Furthermore, thanks to the contribution of the family, some hypotheses regarding the possible modes of transmission of this pathogen between children and pets were outlined.

## 2. Materials and Methods

### 2.1. Microbiological Methods for Bacterial Identification and Serotyping

#### 2.1.1. Sample Collection

The local health authority collected faecal samples from four dogs (D1, D2, D3, and D4) and one domestic rabbit (D5) hosted in the house during the epidemiological investigation, undertaken in order to find the origin of the pathogen which affected Child A.

#### 2.1.2. Identification and Isolation

The identification and isolation of *Salmonella* were tested by cultural examination according to the OIE method [9] at IZSLT UOT (Territorial Operative Unit) Tuscany North (Pisa). After pre-enrichment of the samples in Buffered Peptone Water, selective enrichment was performed in modified semi-solid Rappaport-Vassiliadis (MSRV) Agar, and inoculation of the colonies in Xylose-Lysine-Desoxycholate (XLD) Agar and Shigella Salmonella Agar plates. All the available strains were sent to the Enteropathogenic bacteria Regional Reference Center (CREP) laboratory of IZLST (Istituto Zooprofilattico Sperimentale del Lazio e della Toscana “M. Aleandri”) according to the prescribed sample analysis flow described in Russini et al. [10] in order to perform serotyping and molecular analysis.

#### 2.1.3. Human Strains Collection

The positivity and isolation of *Salmonella* spp. from the stool sample of Child A were documented by the private laboratory which performed the analysis, but the strain was disposed of before further characterization. The *Salmonella* spp. strain from the faecal sample of Child B was isolated by a local public sanitary centre (AUSL Tuscany N-W) and identified using VITEK/MS Maldi-Tof (bioMérieux SA, Marcy-l’Etoile, France). The available strain was brought to the CREP laboratory of IZLST.

#### 2.1.4. Serotyping

Serotyping was performed according to ISO/TR 6579–3:2014 by seroagglutination using antiserum for *Salmonella* (Sifin Diagnostics GmbH, Berlin, Germany; SSI Diagnostica A/S, Hillerød, Denmark; Bio-Rad, CA, USA).

#### 2.1.5. Antimicrobial Resistance

Antimicrobial susceptibility testing (AST) was performed at the National Reference Laboratory for Antimicrobial Resistance (NRL-AR), Department of General Diagnostics, IZSLT, through minimum inhibitory concentration (MIC) determination by broth microdilution using the EU consensus 96-well microtiter plates (Trek Diagnostic Systems, Westlake, OH, USA). The results were interpreted according to the European Committee on Antimicrobial Susceptibility Testing (EUCAST; http://www.eucast.org, accessed on 28 April 2022) epidemiological cut-offs and clinical breakpoints (when available). The following drugs were tested: ampicillin, cefotaxime, ceftazidime, meropenem, azithromycin chloramphenicol, nalidixic acid, ciprofloxacin, colistin, gentamicin, sulfa-methoxazole, tetracycline, tigecycline, and trimethoprim. *E. coli* ATCC 25922 was used as a quality control strain.

### 2.2. Data Collection of Epidemiological Investigation

A questionnaire was prepared for carrying out the epidemiological enquiry. Due to the young age of the patients, the mother of the children filled out the questionnaire. The questionnaire was administered in two parts: the first at the beginning of the enquiry and the latter after results were obtained and the literature was examined. The questions concerned the habits of the family and the pets in the household, the feeding habits of the dogs, and the details of the symptoms and medical treatment undergone.

In addition, written informed consent was signed by the parent for the human cases involved in this work.

### 2.3. Whole-Genome Sequencing and In Silico Analysis

To perform the whole-genome sequencing analysis, genomic DNA was extracted with the automatic extraction system, QIAsymphony (Qiagen, Hilden, Germany). Libraries were prepared using Nextera XT DNA Library Prep and pair-end (2 × 300 bp) run with a MiSeq sequencer (Illumina, CA, USA). Raw reads can be found in the Sequence Read Archive (SRA) at the GenBank database (NCBI) under the BioProject PRJNA901880, BioSamples from SRR22298131 to SRR22298134.

The quality of raw reads was assessed with Fast QC (v0.11.5) [11], and low-quality reads and adapters were trimmed using Trimmomatic (v0.39) [12] before any analysis was done using the following quality filter: minimum quality of Q30, a window size of 10 with Q20 as the average quality, and a minimum length read of 50 bp. The high-quality reads were de novo assembled into contigs using SPAdes (v3.13.0) [13] with the careful option on, draft assemblies were improved using Pilon (v1.23) [14], and contigs shorter than 500 bp were removed [15]. The assembly quality was assessed with QUAST (v5.0.2) [16].

The antigenic formula was deduced in silico by using SeqSero2 v1.1.0 [17,18] on the platform of Deng Laboratory (available at denglab.info/SeqSero2, accessed on 8 April 2022). In silico subtyping was performed with MLST (v2.11) [19] that used the classic scheme for *Salmonella* spp. of seven housekeeping genes (*aroC*, *dnaN*, *hemD*, *hisD*, *purE*, *sucA*, *thrA*) described by Kidgell et al. in 2002 [20]. The cgMLST analyses were performed using chewBBACA (v2.8.5) [21] based on the Enterobase cgMLST v2 scheme [22,23]. The cgSTs were assigned by cgMLSTFinder v1.2 [24] based on the Enterobase database [25] available on the platform of Center for Genomic Epidemiology (https://cge.food.dtu.dk/services/cgMLSTFinder/, accessed on 8 April 2022). Minimum spanning tree was generated using the MSTreeV2 algorithm in the GrapeTree (v1.5.0) software [26].

The SNPs analysis was performed with the pipeline CSI phylogeny (v1.4) [27], attainable from the CGE, using trimmed reads. The SNPs were filtered according to several parameters: a minimum distance of 10 bp between each SNP, a minimum of 10 × depth and 10% of the breadth coverage, a mapping quality above 30, and SNP quality higher than 25.

The presence of virulence genes was assessed using the tool VirulenceFinder v2.0.3 [28] against the Virulence Factors of Pathogenic Bacteria (VFDB) database of core virulence genes [29]. The identification of antimicrobial resistance genes of each strain was assessed from the assemblies using BLAST v2.11 [30] against using the ResFinder database [31] (https://bitbucket.org/genomicepidemiology/resfinder_db/src/master/, accessed on 8 April 2022).

The presence of plasmids was assessed using PlasmidFinder v2.0 [32] accessible from the CGE compared with the Enterobacteriales database (curated by Henrik Hasman and Alessandra Carattoli). The standard quality filters were applied: 95% as the threshold for the minimum percentage of identity and 60% as the minimum percentage of coverage.

## 3. Results

### 3.1. Case Presentation

At the beginning of 2021, a three-year-old child (Child A) manifested gastroenteric symptoms with abdominal pain, fever, frequent diarrhea discharges, and vomiting. The child was hospitalized and recovered after the administration of a physiological infusion, paracetamol, probiotics, and mineral salts. The faecal sample tested three days after the onset of the symptoms resulted positive for *Salmonella* spp., but no further analysis could be performed due to the unavailability of the isolate.

Microbiological analysis was also performed on faecal samples obtained from the asymptomatic mother and the other seven-year-old child (Child B) to assess the spread extent of the contamination in the family. The sampling was carried out nine days after the onset of Child A’s symptoms. The sample of Child B tested positive for *Salmonella* spp., while the mother’s sample (C) tested negative. The day after the sampling, Child B presented similar symptoms, such as mild fever, diarrhea, and abdominal pain in a lighter form. He was treated with mineral supplements and lactic ferments.

Two weeks after the diagnosis from Child A, the local health authority arranged for the sampling of all domestic animals’ faeces living in the same household. Stool samples were drawn from four dogs (D1, D2, D3, and D4) and one rabbit (D5) that lived mainly in the courtyard but could also access the house, with frequent and close contact with the family. The dogs had not been treated with antimicrobials in the previous year, according to the children’s mother. The declared food habits of the dogs included dry food and human meal leftovers. The animals were fed in the courtyard, and children were only occasionally in contact with the animals’ feed, but had free access to the bowls. The timeline of events is reported in Figure 1.

The samples from three dogs (D1, D2, D3) tested positive for *Salmonella* spp., while the fourth dog (D4) and the rabbit (D5) tested negative.

No evidence was collected regarding the consumption of suspicious foods or feeds, and in pursuing the investigation, the local health authority did not sample any household foods or feeds. The strains isolated from the second child and the three dogs were sent to the CREP laboratory of IZLST for further investigations.

### 3.2. Strains Characterization

The clinical isolate and the three veterinarian ones were serotyped and identified as *S. enterica* subsp. *enterica* serovar Typhimurium in its monophasic variant with the antigenic formula 4,5,12:i:- O:4 (B) for all the isolates except for the dog D3 (4,12:i:- O:4 (B)).

All isolates showed the same antimicrobial resistance (AMR) phenotype, displaying microbiological and clinical resistance to ampicillin (MIC ≥ 64 mg/L) and microbiological resistance to sulfamethoxazole (MIC ≥ 1024 mg/L) and tetracycline (MIC > 64 mg/L).

All the involved strains were sequenced and identified as *Salmonella enterica* subsp. *enterica* serovar Typhimurium in its monophasic variant with the genomic predicted serotype 4:i:-. All the strains, after assembled in contigs, were assessed to belong to the sequence type (ST) 34.

The strains isolated from the dogs were sent to the Istituto Zooprofilattico Sperimentale delle Venezie (IZSVe) in accordance with the provisions of the Enter-Vet surveillance network (see Discussion). The network laboratories verified the belonging to MLVA profile 3–12–16-NA-0211 for all the isolates.

The analysis of the cgMLST was performed to evaluate the allele differences to assess the degree of relatedness. The results showed that the strains were tightly related and belonged to the same cgST (cgST250473); the isolate from the child had one allelic distance from the three dogs’ strains, which differed from each other for two allelic distances (Appendix A). The SNPs analysis showed that no SNPs were detected between isolates. The results confirmed the presence of the unique cluster of *S.* Typhimurium var. monophasic that infected the three dogs (D1, D2, D3) and Child B.

The phenotypic resistance patterns were confirmed by the presence of the corresponding AMR genes: all isolates harboured the sulfamethoxazole resistance gene *sul2* [33], the tetracycline resistance gene *tetB* [34], and the ampicillin resistant gene *blaTEM-1B* [35]. The isolates shared the same core virulence genes (145/168), except for the samples from D3, which had four fewer (141/168). The strains carried virulence genes *invA*, *fimA,* and *spiC*, which are important to promoting virulence, involved in the invasion of host epithelial cells (*invA*), and contribute to colonization of the epithelium cells (*fimA*) [36]. The presence of one plasmid was assessed and identified as the ColpVC plasmid (about 2590bp), carrying replication origin and hypothetical protein genes [37]. ColpVC plasmid is not known to be related to resistance factors or other benefits to the bacterial host if present alone. Otherwise, some evidence links this plasmid, concurrent to other Col-like plasmids, to a higher conjugation rate and an increased fitness [38].

## 4. Discussion

Companion animals improve human well-being because of psychological support, friendship, and promoting good health practices. In particular, dogs have a positive impact on the development of children: children with a pet dog present lower anxiety scores than children without pet dogs. Attachment to pet dogs and pet cats during childhood is negatively associated with the risk of mental health disorders in adolescence [39]. However, dogs and cats (together with many other pet animals) can still be a source of many pathogens, including viruses, bacteria, and parasites, accounting for approximately 1% of the morbidity of human salmonellosis per year [5]. The risk of contracting diseases from pet ownership can be reduced with appropriate preventive measures, but it remains not null, especially in front of the manifestation of specific behaviours (kissing, sleeping in the same bed, sharing food or kitchen utensils) or towards high-risk group members such as children or those with an immunodeficiency disorder status [40].

This work reported a case of an *S.* Typhimurium monophasic outbreak involving two children and three family dogs. The genomic analyses ascertained the presence of a unique cluster, including Child B, whose strain was available, and the three positive dogs (D1, D2, D3), since the 1–2 allelic distance was found in the cgMLST (Enterobase database). The SNPs analysis showed no punctual mutation in the different genomes, a discrepancy that could be due to the minimum quality filter used.

Due to the short interval between the two symptom insurgencies, the resemblance of the symptoms, and the presence of asymptomatic dogs presenting the same strain from Child B, we may hypothesize that Child A was part of the same outbreak, sharing the same etiologic agent.

Different ways of transmission can be hypothesized: a transmission by a common source for children and dogs (through ingestion of human food or manipulation of dog feed), a zoonotic transmission from animals to humans or a transmission from humans to animals.

In the case herein reported, the dogs were fed with dry feed and human food leftovers. This kind of habit minimizes but does not exclude the risk of contracting salmonellosis [41,42]. Negativity for the presence of *Salmonella* in the mother’s stool may support the hypothesis that there was not a common food source. However, several factors must be considered, such as a higher infective dose required for adults [43], the probability of *Salmonella* detection in the stool [44], the non-homogeneous diffusion of contamination in consumed foods [10], or the possibility of contamination of food consumed exclusively by the children.

The feeding of the dogs was carried out in the external area of the house and did not directly involve the children, who, however, had free access to the dogs’ feed and bowls and had direct contact with the dogs inside the household. These factors could favor the alternative hypothesis of considering the dogs as asymptomatic carriers of salmonellosis that spread in the environment.

Even if *Salmonella* prevalence in animals is a discussed topic, relatively little is known about the role of diet in determining the possibility that the animal becomes infected and can disperse the pathogen [45]. Pet animals are sometimes exposed to *Salmonella* through the ingestion of contaminated dry feed [46], bones, and raw meat [47,48]. A large survey in the US (2016) affirmed that 3% of dog owners feed their pets raw products, and raw or cooked human food was purchased for pets by 17% of dog owners, which increases the risk of *Salmonella* spp. infection [49]. Commercial feed, including dry food, as in the case described in our study, lowered the risk of asymptomatic carrying of *Salmonella* spp., but did not exclude it [42]. The prevalence of *Salmonella*-positive sampling units for pet food in Europe was 1.1% [1]. Furthermore, dogs usually act as asymptomatic carriers of *Salmonella* spp. and can spread the pathogen for more than six weeks [50]. Studies carried out in Spain and the UK reported a prevalence of *Salmonella* spp. in dogs of 0.23 to 1.85% [42,50]. The prevalence is highly variable depending on the environment in which the animals live. *Salmonella* isolation rates from stray dogs and shelter dogs have been shown to be higher than those from households [49]. However, a previous study associated cases of infant salmonellosis with the habit of feeding pets in the kitchen, and it reported that pet feed is the main source of infections for dogs [51]. Pets may contaminate the environment and other animals by shedding bacteria in their faeces [52], and some factors could increase the load of *Salmonella* spp. spread in the environment [53]. Indirect routes of *Salmonella* spp. transmission from animals to humans are also possible due to the ability of *Salmonella* to survive in the environment, but the faecal-oral route remains the most common one that leads to human infection [49].

Unfortunately, given the lack of sampling from food, feed, and environment during the investigation, none of these hypotheses could be proved [10].

Concerning the hypothesis of the human-to-animal infection, it is known that children are more susceptible to the disease than healthy adults and may be a spreader for a longer time. In humans, *Salmonella* is excreted in feces after infection for a median of 5 weeks, a period that can be even longer in children [43]. In the case described here, isolation of *Salmonella* in pet feces occurred two weeks after the diagnosis of case A, allowing the possibility of human-to-animal infection.

Given the available data, the directionality of the transmission could not be assessed, and the different hypotheses of infections as animals-to-humans, humans-to-animals, or infections caused by food or feed could be valid.

Transmission of bacterial pathogens can also resolve in the passage of AMR determinants, as in this described outbreak where the bacteria involved were carriers of AMR determinants mediating resistance to three antimicrobial classes ((amino) penicillins, sulphonamides and tetracyclines). Potential routes for antimicrobial resistance factors to humans include direct handling or close contact between infected animals and humans, transmission via contaminated food products, and transmission via contamination of the environment [54]. Reciprocal affectionate behaviours between humans and their companion animals could contribute to transmission of drug-resistant microorganisms. Despite many pet owners perceiving that close contact with animals is unhygienic and sometimes socially sanctioned, they are not aware of the significance of those behaviours as potential drivers of AMR. Because affectionate behaviours are crucial for the relationship between humans and pets, it has been suggested that reducing the number of antimicrobials shared between human and veterinary medicine would be a more feasible way than requesting a change of behaviour from the owners in order to address the issue of AMR transmission [55]. However, the dogs involved in the episode described here had not been treated with antimicrobials in the last year. This would suggest that the strain of *Salmonella* spp. isolated in the dogs’ stools was not promoted by a previous treatment but was contracted from an external source.

The strains within the described outbreak showed a different serological formula, assigned with classical methods of seroagglutination using antiserum for *Salmonella* (samples D1 and D2 4,5,12:i:- versus sample D3 4,12:i:-). Achtman and colleagues in 2012 [56] pointed out that the serotyping nomenclature could be phylogenetically inconsistent, and the MLST is a powerful candidate for the reference classification system for *Salmonella* and can replace serotyping for that purpose. However, MLST does not provide the fine resolution needed for outbreak analysis and short-term epidemiology. The WGS analysis, on the other hand, provides essential information for epidemiological monitoring and provides valuable information on the detailed structure of populations and for the resolution of outbreaks. Thus, the *Salmonella* serotype formula can not be used as a method of discrimination to direct investigations on samples within an outbreak.

The *S.* Typhimurium monophasic ST34 is one of the most frequently reported STs for *S.* Typhimurium. Due to its rapid dissemination worldwide and multidrug resistance, it is considered a raised global concern [57]. In Europe, one of the last large ST34 outbreaks reported was the multi-country outbreak involving at least 445 cases (in two clusters), mainly children, caused by chocolate products [58,59]. Importantly, ST34 is mainly associated with the pattern of resistance to ampicillin, chloramphenicol, streptomycin, sulbactam, and tetracycline [60], even if the attention is focused on the ST34 strains with colistin resistance mechanism (mcr-1) [57,61,62,63] not present in the outbreak herein described. In Italy, the ST34 count is 131 isolates in the public Enterobase database, predominantly from livestock, poultry, food, feed, and humans in the years 2008–2022. Worldwide, the Enterobase database contains both ST34 Typhimurium and its monophasic variant [56], and contains about 24,000 isolates (1992–2022). Looking for the same cgST, we found a human strain from France that perfectly matches the human strain analysed in this study, having no allelic difference in the cgMLST analysis. We do not have any information about this particular strain, and more investigation can be carried out to understand if any epidemiological link could explain this high similarity to the familiar outbreak here described.

In Italy, the management of data regarding infectious diseases is committed to the Infectious Diseases Information System, based on reports from physicians. Non-typhoid salmonellosis, such as those caused by *S.* Typhimurium and its monophasic variant, belong to class II infectious diseases, with the obligation to notify the case to the Local Health Authority within 48 h from diagnosis. Furthermore, in Italy, a special surveillance laboratory system for the control of salmonellosis is maintained by Enter-Net Italia, coordinated by the Italian National Institute of Health (ISS) that collects the strains and epidemiological and microbiological information relating to the isolations of *Salmonella* of human origin [62]. The network collects one out of ten of all the human strains isolated by the laboratories and performs MLVA analysis. Genomic sequencing is not carried out routinary, but only in the case of significant nationwide outbreak events. The isolate of *Salmonella* spp. from samples of veterinary origin (food, animals, environment, and water) are collected by the Enter-Vet surveillance network, coordinated by the National Reference Center for Salmonellosis of the IZSVe, and aims to collect isolates and data at the national level. All the strains belonging to the serovar Enteritidis, Typhimurium and its monophasic variant are collected, in addition to the strains collected in other specific monitoring activities. Additionally, in this case, MLVA analysis is performed [64]. The MLVA 3–12-16-NA-211 profile is not widely diffused on the national territory. From 2019 to today, the Enter-Net network has recorded 31 strains of human origin attributable to this profile, of which 2 in the year 2021 occurred in the regions of Piedmont and Marche (personal communication from Enter-Net Italia Network). On the veterinary side, in Italy from 2020 to date, this profile has been identified in 17 strains, most of which were isolated from pigs and pork-based foods. Strikingly, one of the latter was sampled in a restaurant during a foodborne disease epidemiological investigation. In the future, further sequencing analyses on a selection of these strains could be of support for a correct attribution of the contamination sources.

The outbreak studied shows the difficulty of establishing the first origin of a zoonotic bacterial outbreak in the case of cohabitation between humans and pets. The whole-genome sequencing is confirmed as the elective methodology for *Salmonella* outbreak investigations. Genomic analysis is valuably supported by the matching of data collected by genomic surveillance platforms. Some missing information concerning the case (absence of a structured questionnaire for the family administrated at the moment of the hospitalisation, absence of analysis of the food and the feed leftovers at the time of the investigation, and abstaining of storage and further analysis from the first sample from Child A) highlights the need for a unique procedure for the surveillance bodies.

This work confirms the need to implement and standardize control measures by local health authorities, adopt whole-genome sequencing as the method of choice for the analysis of *Salmonella* outbreaks, and support this analysis with the establishment of platforms for genomic surveillance and the integration of control systems for isolates of human, food, and veterinary origin from a One Health perspective.

## Figures and Tables

**Figure 1 pathogens-11-01500-f001:**
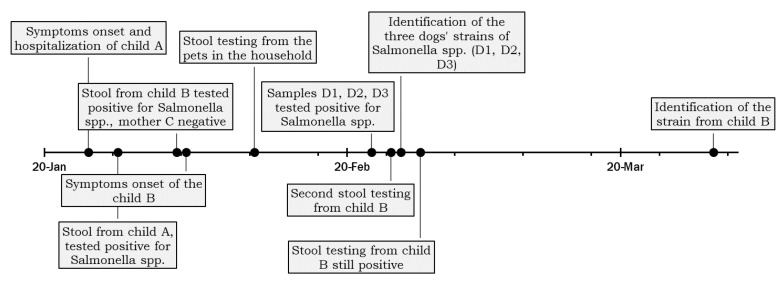
Timeline of the events leading to the detection of the family outbreak described.

## Data Availability

Raw reads can be found in the Sequence Read Archive (SRA) at the GenBank database (NCBI) under the BioProject PRJNA901880, BioSamples from SRR22298131 to SRR22298134.

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
