# Peer review of "A Familiar Outbreak of Monophasic Salmonella serovar Typhimurium (ST34) Involving Three Dogs and Their Owner’s Children"

_pathogens, 2022, doi:10.3390/pathogens11121500_

Round 1

Reviewer 1 Report

In this manuscript, Russini et al described the detection of closely related isolates of monophasic Salmonella Typhimurium ST34 in one/two children and three pet dogs of the same family.  The manuscript is generally well written; the identification of the bacterial isolates, including its sequence type, antigenic assignment and antibiotic resistance profile is supported by the methods described.  The sequencing data, and in particular the allelic type data, should be shown.

The authors argued for importance of data collection in public health approach to infectious disease, and rightfully so; the authors also argued that companions pets (e.g. dogs) can have a direct role in the transmission of Salmonella, this argument, while seldom demonstrated and would be impactful if it could be demonstrated, is not support by the data in its current form.  This is the main weakness of this manuscript as it stands.  While I appropriate that data collection occurred retrospectively and there were practical constraints on what could be done, without convincing data on directionality of transmission (which requires sequential data collection over a time period and extensive environmental sampling, as a minimum) this report shows a family with one child and three dogs infected with the same strain of ST34 and not much more.

Salmonella Typhimurium is a food-borne bacterial pathogen and the family dogs are known to have eaten human food leftover.  A simple explanation (though not the only one) would be that children and dogs acquired the same strain of Salmonella by eating the same, contaminated food.  The authors rationalised that if human food was implicated then Salmonella should be detected in the stool sample from the mother, but it was not.  However consumption of Salmonella-contaminated food doesn't always lead to bacterial detection in the faecal sample.  A controlled human challenge model of Salmonella Typhi (closely related to S. Typhimurium) found that approximately 60% of experimentally infected human volunteers exhibit clinical symptoms within 14 days and less than half of them shed Salmonella in the faeces (Darton et al, PMID 27533046).  Children are also more susceptible to S. Typhimurium than healthy adults.  Conversely, ST34 was detected in three out of four of the dogs in the family, suggesting that penetrance is incomplete in dogs too.  

In addition I also have a few minor comments:

- it is stated in the abstract in and in the introduction that salmonellosis is the second most reported bacterial zoonosis in humans; this statement is not supported by the reference;

- the report presents a case study but detection of Salmonella in the same family is not an outbreak unless surrounding cases of the same strain has been detected, and if so should be included and discussed;

- line 168, it's not clear what the "local competent authority" means;

- the manuscript mentions Fig 1 but the only figure is labelled as Fig 2;

- line 199, it's not clear what "the strain brings virulence genes" means;

- line 209-210 assumes child A had the same strain of Salmonella but without data is not substantiated.

Author Response

In this manuscript, Russini et al described the detection of closely related isolates of monophasic Salmonella Typhimurium ST34 in one/two children and three pet dogs of the same family.  The manuscript is generally well written; the identification of the bacterial isolates, including its sequence type, antigenic assignment and antibiotic resistance profile is supported by the methods described.  The sequencing data, and in particular the allelic type data, should be shown.

The authors argued for importance of data collection in public health approach to infectious disease, and rightfully so; the authors also argued that companions pets (e.g. dogs) can have a direct role in the transmission of Salmonella, this argument, while seldom demonstrated and would be impactful if it could be demonstrated, is not support by the data in its current form.  This is the main weakness of this manuscript as it stands.  While I appropriate that data collection occurred retrospectively and there were practical constraints on what could be done, without convincing data on directionality of transmission (which requires sequential data collection over a time period and extensive environmental sampling, as a minimum) this report shows a family with one child and three dogs infected with the same strain of ST34 and not much more.

Salmonella Typhimurium is a food-borne bacterial pathogen and the family dogs are known to have eaten human food leftover.  A simple explanation (though not the only one) would be that children and dogs acquired the same strain of Salmonella by eating the same, contaminated food.  The authors rationalised that if human food was implicated then Salmonella should be detected in the stool sample from the mother, but it was not.  However consumption of Salmonella-contaminated food doesn't always lead to bacterial detection in the faecal sample.  A controlled human challenge model of Salmonella Typhi (closely related to S. Typhimurium) found that approximately 60% of experimentally infected human volunteers exhibit clinical symptoms within 14 days and less than half of them shed Salmonella in the faeces (Darton et al, PMID 27533046).  Children are also more susceptible to S. Typhimurium than healthy adults.  Conversely, ST34 was detected in three out of four of the dogs in the family, suggesting that penetrance is incomplete in dogs too. 

We thank the reviewer for these valuable suggestions. We have extensively reformulated the discussion by including and better describing the suggested hypotheses. We are aware of the impossibility of demonstrating the directionality of the transmission through the analysis of the data available according to a retrospective approach. We believe that this type of demonstration would be guaranteed only in the event of a detection of a contaminated food not yet handled among the ones investigated. But in the conduction of an epidemiological investigation this kind of finding is very rare.

However, we consider important to disclose works of this kind also to provide information useful for standardization of the actions undertaken by the local health authorities (for example: creation of guidelines which also recommend environmental sampling), to reaffirm the importance of adopting whole genome sequencing as a routine method in epidemiological and source attribution surveys and to promote the adoption of salmonella genomic data exchange platforms at a national level.

In addition I also have a few minor comments:

- it is stated in the abstract in and in the introduction that salmonellosis is the second most reported bacterial zoonosis in humans; this statement is not supported by the reference;

We cited the European Union One Health 2020 Zoonoses Report to support this statement (line 44). In fact it is reported that in 2020, the first and second most reported zoonoses in humans were campylobacteriosis and salmonellosis, respectively.

- the report presents a case study but detection of Salmonella in the same family is not an outbreak unless surrounding cases of the same strain has been detected, and if so should be included and discussed;

We used the term outbreak for the familiar episode following the CDC definition of an unexpected number of cases in a limited geographic area, as indicated form the following sources:

For ECDC: “An outbreak is defined as two or more cases where the onset of illness is closely linked in time (weeks rather than months) and in space, where there is suspicion of, or evidence of, a common source of infection, with or without microbiological support (i.e. common spatial location of cases from travel history)”.

For EFSA: “An excess of disease cases compared to what would be normally expected in a population. An outbreak may occur in a restricted geographical area, or may extend over several countries. It may last for a few days or weeks, or for several years”.

For HPSC: “An outbreak of infection or foodborne illness may be defined as two or more linked cases of the same illness or the situation where the observed number of cases exceeds the expected number, or a single case of disease caused by a significant pathogen (e.g. diphtheria or viral haemorrhagic fever). Outbreaks may be confined to some of the members of one family or may be more widespread and involve cases either locally, nationally or internationally”.

- line 168, it's not clear what the "local competent authority" means;

We thank the reviewer for the indication. The term was fixed in “local health authority”.

- the manuscript mentions Fig 1 but the only figure is labelled as Fig 2;

We thank the reviewer for the indication. We fixed in Fig. 1

- line 199, it's not clear what "the strain brings virulence genes" means;

We thank the reviewer for the suggestion. We fixed it in “the strains carry virulence genes” (line 210)

- line 209-210 assumes child A had the same strain of Salmonella but without data is not substantiated.

We thank the reviewer for making us understand that this statement should be explicitly reported in the text. We fixed the statement in this way:

“Due to the short interval between the two symptom insurgencies, the resemblance of the symptoms, the presence of asymptomatic dogs presenting the same strain from child B, we may hypothesize that child A was part of the same outbreak, sharing the same etiologic agent” (line 237-240)

Reviewer 2 Report

In the present manuscript, the authors reported an outbreak of monophasic Salmonella Typhimurium sequence type 34 in a family affecting two children and involving their three dogs as carriers in central Italy. In general, the manuscript is well written and the study design is appropriate. I only have a few minor comments to be addressed:

Genes names should be in italics.

Line 35: “diseases” change to “disease”.

Line 88: Please define initials “IZLST”.

Line 126: The authors should include the minimum spanning tree figure in the manuscript or as supplementary material.

Line 172: “Figure 2” change to “Figure 1”. Please check the order of the third and fourth text boxes. 

Line 202: The authors should discuss the presence of ColpVC plasmid in the manuscript.

Line 267: The authors should cite papers related to the identified resistances.

Line 300: “even if the major concern”. Does not make sense, please rephrase.

Author Response

In the present manuscript, the authors reported an outbreak of monophasic Salmonella Typhimurium sequence type 34 in a family affecting two children and involving their three dogs as carriers in central Italy. In general, the manuscript is well written and the study design is appropriate. I only have a few minor comments to be addressed:

Genes names should be in italics.

We thank the reviewer. We corrected according to the indication.

Line 35: “diseases” change to “disease”.

We thank the reviewer. We corrected according to the indication.

Line 88: Please define initials “IZLST”.

We thank the reviewer. The full name of our institution has been added according to the indication.

Line 126: The authors should include the minimum spanning tree figure in the manuscript or as supplementary material.

We thank the reviewer. The figure has been added in the supplementary material. We also added the distance matrix of cgMLST.

Line 172: “Figure 2” change to “Figure 1”. Please check the order of the third and fourth text boxes.

We thank the reviewer. We fixed in Fig. 1.

The order of the text boxes is correct: the onset of the symptoms happened a day after the stool test. The mother asked for sampling the stool of herself (case C) and the child (B) after the occurrence of the disease in the child (A), and before the reported onset of the symptoms of case B.

Line 202: The authors should discuss the presence of ColpVC plasmid in the manuscript.

We thank the reviewer. We reported information about this plasmid (lines 213-217).

Line 267: The authors should cite papers related to the identified resistances.

We thank the reviewer. The required references were added to the paper.

Line 300: “even if the major concern”. Does not make sense, please rephrase.

We thank the reviewer, we rephrased as follow:

“even if the attention is focused on the ST34 strains with colistin resistance mechanism (mcr-1)” line 326

Reviewer 3 Report

the methods of microbiological methods for Bacterial Identification and Serotyping were not displayed well, please divide this into subheadings including sampling, isolation, serotyping, and antimicrobial susceptibility testing, and each sector should include details of the method and references to be more organized for the reader.

please provide the accession number of the sequenced strain in the result section.

Author Response

the methods of microbiological methods for Bacterial Identification and Serotyping were not displayed well, please divide this into subheadings including sampling, isolation, serotyping, and antimicrobial susceptibility testing, and each sector should include details of the method and references to be more organized for the reader.

The method section has been extensively revised and divided in subheadings according the reviewer suggestion

please provide the accession number of the sequenced strain in the result section.

Accession number of strain sequences have been added in the “Results” section and in the “Data Availability Statement”

Round 2

Reviewer 3 Report

The authors considered the reviewer comments